# OpenReview forum: "GOVSIM-ELECT: Elections in AI Societies"
_colmweb.org/COLM/2025/Workshop/Social_Sim — Social Sim'25_

### Official Review · Reviewer_KQRc · 2025-07-03
**Violation of Double-Blind Review Policy in Submission**

**Rating:** 1
**Overall Assessment:** 1
**Confidence:** 1

**Review:**

N/A

**Comments Suggestions And Typos:**

N/A

**Paper Summary:**

Dear Area Chairs,

I noticed that this submission violates the double-blind review policy by including the authors' names on the paper PDF. I wanted to bring this to your attention so you can determine how best to handle the situation.

**Relevance:**

1

**Summary Of Strengths:**

N/A

**Summary Of Weaknesses:**

N/A

---

### Official Review · Reviewer_9Ysk · 2025-07-10
**Use of LLM-based agents to simulate elections**

**Rating:** 4
**Overall Assessment:** 2
**Confidence:** 5

**Review:**

The work is quite original, even though its quality and significance are somewhat questionable. Regarding the quality, the analysis is at times quite confused (broken references to the Appendix, lack of readability of the figures, e.g., legend in Fig. 2c, and even possibly description of the wrong figure - comments to Fig. 3c do not reflect its content, and lack of description of their metrics, e.g., they talk about resource management performance, but that's not at all defined, nor it is clear from the model pipeline). Regarding the significance, I appreciate the comparison across different LLMs, however, very little is known about the full model pipeline methodology (e.g. model temperature?), therefore, it is difficult to judge how generalizable the results are. Furthermore, it is unclear why they chose only 13 agents, but that seems to be a strong limitation of the study.

**Comments Suggestions And Typos:**

As mentioned above, there are a number of mistakes:
- there are broken references to the appendix. Also, the content of the Appendix could be framed.
- Figures and comments do not always match, e.g., Fig. 3c.
- Quality of the figures could be improved, e.g., legend of Fig 2c, and 3c.
- The connection between the voting and the resource allocations is completely overlooked. Furthermore, comments related to the resource distribution performances are not supported by plots/results.

**Ethical Concerns:**

Authors' names are not masked. Also, the length exceeds the limit of 9 pages.

**Paper Summary:**

Building on previous work, the authors propose a study of a multi-agent LLM-based system where they simulate multi-round elections. For their study, they compare 5 different open-weight models. Ultimately, they are especially interested in the role of 4 different leaders (with/without reasoning and using clear/verbose statements).
The results vary across the different models, although overall they show a preference towards reasoning and mostly clear statements.

**Relevance:**

4

**Summary Of Strengths:**

Comparison between different LLMs.

**Summary Of Weaknesses:**

- the analysis is confused
- problem setup is not clear, lack of details in the methodology
- case study seems too small.

---

### Meta-Review · Area_Chair_Kt4Z · 2025-07-21

**Recommendation:** Accept

**Metareview:**

Please refer to the feedback from  Reviewer 9Ysk.

Please note the paper was restricted to 9 papers for the main text, and should be double-blind.